# Crash Risk Predictors in Older Drivers: A Cross-Sectional Study Based on a Driving Simulator and Machine Learning Algorithms

**DOI:** 10.3390/ijerph20054212

**Published:** 2023-02-27

**Authors:** Vanderlei Carneiro Silva, Aluane Silva Dias, Julia Maria D’Andréa Greve, Catherine L. Davis, André Luiz de Seixas Soares, Guilherme Carlos Brech, Sérgio Ayama, Wilson Jacob-Filho, Alexandre Leopold Busse, Maria Eugênia Mayr de Biase, Alexandra Carolina Canonica, Angelica Castilho Alonso

**Affiliations:** 1Laboratory for the Study of Movement, Department of Orthopedics and Traumatology, School of Medicine, University of São Paulo, São Paulo 05403-010, Brazil; 2Graduate Program in Aging Science, São Judas Tadeu University (USJT), São Paulo 03166-000, Brazil; 3Georgia Prevention Institute, Medical College of Georgia, Augusta University, Augusta, GA 30901, USA

**Keywords:** safe driving, older drivers, crash risk, clustering analysis, machine learning

## Abstract

The ability to drive depends on the motor, visual, and cognitive functions, which are necessary to integrate information and respond appropriately to different situations that occur in traffic. The study aimed to evaluate older drivers in a driving simulator and identify motor, cognitive and visual variables that interfere with safe driving through a cluster analysis, and identify the main predictors of traffic crashes. We analyzed the data of older drivers (n = 100, mean age of 72.5 ± 5.7 years) recruited in a hospital in São Paulo, Brazil. The assessments were divided into three domains: motor, visual, and cognitive. The K-Means algorithm was used to identify clusters of individuals with similar characteristics that may be associated with the risk of a traffic crash. The Random Forest algorithm was used to predict road crash in older drivers and identify the predictors (main risk factors) related to the outcome (number of crashes). The analysis identified two clusters, one with 59 participants and another with 41 drivers. There were no differences in the mean of crashes (1.7 vs. 1.8) and infractions (2.6 vs. 2.0) by cluster. However, the drivers allocated in Cluster 1, when compared to Cluster 2, had higher age, driving time, and braking time (*p* < 0.05). The random forest performed well (r = 0.98, R^2^ = 0.81) in predicting road crash. Advanced age and the functional reach test were the factors representing the highest risk of road crash. There were no differences in the number of crashes and infractions per cluster. However, the Random Forest model performed well in predicting the number of crashes.

## 1. Introduction

The number of older drivers has been increasing, and this can be considered a consequence of the rise in life expectancy. In addition, there is also a high number of traumas and crashes involving this population, which is three times more likely to be involved in crashes than younger adults, and higher risks of hospitalizations and motor impairment inherent to hospitalization [1,2].

By 2030, in the United States, there will be more than 70 million people aged 65 and older, and approximately 85–90% of them will be licensed to drive and, for the first time in history, there is a need to plan how long they will continue to drive safely, just as they project financial retirement [3].

Older adults are among the most careful drivers on the road and often reduce the risk of injuries by wearing seat belts, not drinking alcohol, and because they respect speed limits, but they also are more likely to die in a crash due to age-related frailty. In 2018, 6907 people aged 65 and over died in a road crash, representing 19% of all fatalities [4]. Age-related impairments in functional abilities needed for safe driving may increase crash and injury risk, reducing mobility, autonomy, and quality of life [5]. 

Driving is a complex task that involves the integration of motor, cognitive, and visual skills. The increase in age is related to changes in these capabilities that impact vehicle direction, which can contribute to the risk of a crash in older drivers. Age-related changes in visual function (mainly reduced visual acuity and visual field loss) have been investigated as risk factors for crashes among older adults [6]. According to Wang et al. [5], older drivers have more difficulty rapidly assessing and responding to roadway changes. Impaired contrast sensitivity was associated with a higher number of crashes and an increased severity of crashes [7]. In addition, it is known that the natural cognitive decline of aging can affect driving, mainly due to attention deficit, impairments in memory, spatial and visual notions, and a decline in attention, which are fundamental characteristics and directly impact safety while driving a vehicle [7]. The visuospatial skills involve the mental representations of the shape of objects and their locations and cognitive processes of transforming objects and movement through space. In the study by Anderson et al. [8], drivers who had accidents on the driving simulator showed worse performance in the composite measures of cognitive function, specifically in visuospatial skills and attention. Tinella et al. [9] evidenced significant results for the effect of global cognitive functioning on perceptual speed through the full mediation of mental rotation and perspective-taking skills. In another study by Tinella et al. [10], the results suggest the specific contribution of spatial mental transformation skills in the execution of complex behaviors connected to the fitness to drive. The physical condition of older adults explains the high risk of injuries and deaths. Alonso et al. [11] affirm that muscle strength, postural balance, and cognition are associated with braking time and may affect driving performance in older adults. Age-related sensorimotor impairments also result in poor stepping reactions, altered patterns of movement coordination, and slower-reaching reactions in response to postural disturbances. Lower limb strength and control may be important for vehicle control, particularly in coordinating and adjusting the accelerator, brake, and clutch pedals. A test battery that evaluation of older drivers, ensuring that those who are unsafe to drive are identified is very important [6,11]. However, there is still no consensus on which battery is the best, and what has been sought are tested with high predictive values with fewer variables.

Health problems are the main reasons found that limit the driving of vehicles in this population. However, socio-demographic characteristics also have influences such as age, gender, educational level, marital status, urban residence, recent hospitalizations, psychic, physical, and degenerative diseases, social issues such as low income and unemployment, trauma due to involvement in collisions, interference of some family member, and medical guidance [12].

The stoppage of driving in this population seems to contribute to a variety of health problems, especially depression. For this reason, health-related consequences should be considered in face of this decision. Identifying risk factors and interventions that ensure mobility and maintenance of social activities are necessary to reduce the potential adverse effects of the aging process, in order to promote health and well-being for the older adults [13].

Machine Learning (ML) is an expanding field because of its high predictive power. Algorithms based on ML can help in tasks that involve data analysis and identify the characteristics of elderly drivers associated with performance when driving a vehicle [14]. We tested the hypothesis that driver’s clusters can be derived by an unsupervised learning algorithm (K-Means) and that the number of road crashes can be predicted by a supervised learning algorithm (random forest) from a data set of motor, visual, and cognitive tests.

The identification of age-related factors, including changes in cognitive, visual, and physical characteristics may help to develop specific interventions that can preserve driving performance during aging, and mitigate unsafe driving actions [11]. For this reason, the present study aimed to evaluate older drivers in a driving simulator and identify motor, cognitive, and visual variables that interfere with safe driving through a cluster analysis and identify the main predictors of traffic crashes.

## 2. Materials and Methods

A cross-sectional analysis was conducted at University São Judas Tadeu in partnership with the Institute of Orthopedics and Traumatology of Clinicals Hospital of Medical School at the University of São Paulo (IOT-HC/FMSUP), at the Laboratory for the Study of Movement. This research was approved by the Research Ethics Committee under number 063/15. All participants were informed about the objectives of the study and those who agreed to participate signed the informed consent form.

### 2.1. Participant Recruitment

This is a study with a convenience sample of 100 older drivers of both genders, aged 60 years old or older, who were recruited at the Laboratory for the Study of Movement, at the IOT-HC/FMSUP. Inclusion criteria were having a valid driver’s license and driving at least two days a week; no significant limitations in movement of the ankle, knee, hip, and cervical joints; not using medications that could alter the ability to drive; absence of vestibular, neurological, and mental diseases; absence of surgeries that can influence the mobility of the spine and limbs. People who were not able to perform the tests on the driving simulator and/or the visual, physical, and cognitive tests were excluded.

### 2.2. Data Collection

All subjects answered a questionnaire with personal information, socio-demographic data, and driving history. They were asked to identify their preferred leg for kicking a ball, which was then considered their dominant leg and the dominant upper limb based on their preferred arm to write. The participants attended one session in the Laboratory for Study of Movement, and they were submitted to the functional tests there, which lasted approximately 1.5 h.

### 2.3. Self-Perception of Disability

Self-perception of disability, based on the Candrive II/Ozcandrive cohort study [15], with 30 questions were created in the following domains: behavior: how to behave in the face of various stimuli; perception: the way of conceptualizing, judging, or qualifying something; cognition: the process of knowing or acquiring knowledge; vision: perception through the eyes; motor: body movements and asking if the participant had difficulty or not—it was calculated with one point for each statement—“I have difficulty…”.

### 2.4. Driving Simulator Test 

The dependent variable evaluated was the number of crashes. The virtual environment was generated by “Car-Simulator Trainer—Type F12PT” (Foerst GMBh), which simulates a vehicle equipped with a steering wheel, speed dial, brake, accelerator, clutch pedals, gear stick, seat, seatbelt, and headlights, and the driving route was visible on three 42″ LCD TV monitors. The participants were instructed to adjust the seat, seatbelt, and rearview mirror as he/she would ordinarily do upon sitting down in the driving simulator. They were familiarized with a virtual scenario consisting of a highway without traffic. To assess the safe driving of drivers, the risk situation test was chosen, with eight unpredictable situations causing crashes, similarly to crashes that occur on the streets and avenues of large cities, where, during the route, a risky situation appeared and forced the driver to take action to avoid the crash. The test scenario consisted of a city with multi-lane streets, road signs, traffic lights, and traffic. In each risky situation, there was a different condition from the other: three situations in which a pedestrian could be run over, a cyclist, and four others in which there could be car collisions. Another test called “braking time” was chosen, in which during the ride, the word “stop” appears randomly five (5) times on the screen, and the subjects were required to brake the car immediately, as fast as they could, and the braking time was measured by the car simulator (model “Trainer” Type F12PT FOERST^®^, Wiehl, Germany). The equipment used provided information on the number of crashes and violations committed by the driver during the route [11,16].

### 2.5. Measures

#### 2.5.1. Motor Domain

##### Handgrip Strength 

Maximal handgrip strength was determined with a hydraulic hand dynamometer (model Jamar^®^ by JLW INSTRUMENTS^®^, Chicago, IL, USA), with the subject seated and with the arms parallel to the body, shoulder adducted, elbow flexed at 90°, and forearm and wrist in the neutral position. Three trials were performed on the dominant and non-dominant hands, with a one-minute interval between trials. The mean value was used for the analyses [17].

##### Functional Reach Test

Assesses the ability of the trunk to move forward within the limits of stability. The individuals lean forward starting from the orthostatic position, positioned perpendicular to the wall, with 90° flexion of the shoulder, elbows extended, and heels together. The data used is the distance covered by the third metacarpal along the horizontal axis measured with a measuring tape. Three attempts were performed, and the average was calculated [18]. 

##### Plantar Flexor Muscle Strength

Maximum dynamic strength of the plantar flexor muscles of the dominant and non-dominant lower limbs were measured using an isokinetic dynamometer (Biodex System 2, Shirley, NY, USA). Subjects were placed in a seated position, with support in the distal region of the thighs, and the soles of the feet resting on a rigid plate. The axis of motion of the ankle joint was aligned with the mechanical axis of the dynamometer, and the knee was kept at 30° of flexion. The subjects were strapped in by two belts across the chest and one across the pelvis, and with velcro strips on the distal part of the thigh and area of the metatarsals in the dorsal region of the foot. Three submaximal attempts were made to familiarize the subject with the procedures, and then two sets of five maximal dynamic repetitions were conducted at 30°/s, with a one-minute interval between sets. Only the second set of values was used for the data analyses. Verbal encouragement was given throughout the tests to motivate the participants to develop the maximum torque during each repetition [11].

##### Dynamic Balance 

Dynamic balance was evaluated with the Timed Up and Go Test (TUG test), comprised of mobility, transfers, and gait, and is associated with strength, agility, and postural balance. The TUG test measures the time required for an individual to get up from a chair to a standing position, walk three meters at a normal walking speed, return to the chair, and sit back down. Additionally, subjects performed the TUG test with a dual-task (“Time Up and Go Cognitive”), which paired the motor activities with a verbal task, and in which the individuals were asked to name animals [16].

##### Articular Amplitude

Rotation of the cervical spine (0°–55°). Individual sitting with the head and neck in an anatomical position. The side to be evaluated is rotated. The goniometer’s fixed arm is positioned in the sagittal suture (center of the head) and the mobile arm must be placed in the sagittal suture at the end of the movement. Shoulder flexion (0–180°) with the volunteer seated, with the arms by the body and elbows extended, the stationary arm of the goniometer is placed along the middle axillary line of the trunk, pointing to the greater trochanter of the femur, and the moving arm of the goniometer on the lateral surface of the humeral body facing the lateral epicondyle of the wrist [19]. 

#### 2.5.2. Visual Domain

Visual acuity assessment (monocular and binocular) used the Snellen optometric scale, which consists of a set of letters, which become progressively smaller from top to bottom. On this scale, normal visual acuity is called 20/20 [20]. For each correct answer, half of a point per letter was considered. 

Visual campimetry—measurement of the unilateral 90° and 180° temporal visual field for both eyes. For this test, the RZ 2000 (equipment by Raizamed^®^, São Paulo, Brazil) was used [20].

#### 2.5.3. Cognitive Domain

##### Montreal Cognitive Assessment (MoCA)

An instrument developed to screen for mild cognitive impairment and access different cognitive domains: attention and concentration, executive functions, memory, language, visual-constructive skills, conceptualization, calculation, and guidance. The total score is 30 points. A score of 26 or more is considered normal [21].

##### Trail Making Test (Trails B)

Part B is a test that assesses aspects of sustained and alternating attention, mental flexibility, visual processing speed, motor function, and ability to search by visual scanning [22]. 

### 2.6. Statistical Analysis

A descriptive analysis of continuous variables was presented as the mean and standard deviation, while categorical variables were presented as the frequency and proportion. The Kolmogorov–Smirnov test was used to verify whether the continuous variables had a normal distribution, and histograms were examined. Comparisons of the mean values of continuous variables by cluster were performed using Student’s *t*-tests. 

The following packages were installed and executed: Cluster and Facto Extra. Analyses were also performed in R software, version 4.0.2. The K-means clustering algorithm was used to distribute the participants into groups based on their characteristics [23]. K-means clustering is one of the most popular cluster algorithms. The data were converted to z-scores and inputted into the algorithm. We retained two clusters considering homogeneity in the derived groups, and the balance between classes [24]. The group’s interpretability was examined to confirm the final number of clusters and if a group was sufficiently large for adequate statistical power, that is, at least 10% of the total sample. The clustering distance measurements were carried out using Euclidean distances [14].

The random forest (RF) algorithm is based on the ensemble strategy. It provides diversity by using the concept of random redistribution of the data. The algorithm generates several decision trees, each trained with a random distribution. The RF algorithm was implemented in the R software (random forest function), in which 70% of the data were used for training and 30% for testing. The analysis was performed using the following established hyperparameters: number of trees (ntrees) = 500, minimum size of terminal nodes (nodesize) = 5, number of variables randomly sampled as candidates at each split (mtry) = 9, the importance of predictors (importance) = True, forest retained (keep.forest) = True, number of folds in the cross-validation (cv.fold) = 10, and seed for reproducibility (seed) = 123. The results were graphically expressed through regression, and the final result was the mean of all results of the regression tree. The model performance was assessed by Pearson’s correlation coefficient between the observed and predicted values. The coefficient of determination (R^2^), and the mean absolute error (MAE).

## 3. Results

### 3.1. Descriptive Analyses

Table 1 shows the demographic characteristics and driving-related data of the subjects in this study. The sample consisted of older adult drivers with a mean age of 73 years, mean 12 years of education, and 48 years of driving experience; 50 (50%) female and 50 (50%) male. In the sample, the mean number of crashes during the tests carried out in the driving simulator was 1.8, while the mean number of infractions was 2.4.

### 3.2. Clusters

Figure 1 shows the clusters with the K-Means algorithm. Therefore, two groups were used to carry out the clustering. The two dimensions explain 20.4% of the variability of the analyzed data.

### 3.3. Characteristics of Older Adult Drivers by Cluster

Table 2 shows the sociodemographic data and motor, visual, and cognitive characteristics by cluster. From the original data set (n = 100), the following two major clusters were derived: Cluster 1 with 59 (59%) participants, 29 (49%) male and 30 (51%) female; Cluster 2 with 41 (41%) participants, 21 (51%) male and 20 (49%) female. There were no differences in the total number of crashes and infractions between the clusters. However, the drivers allocated in cluster 1, when compared to cluster 2, had a higher age, driving time, braking time, and less education. In the motor domain, cluster 1 presented lower handgrip strength (on both sides), shorter functional reach, less shoulder flexion amplitude, and less cervical rotation range of motion. Cluster 1 also presented a long time to perform the TUG test. Regarding visual capacity, the drivers in cluster 1, compared to cluster 2, presented lower values in the right, left, and binocular Snelling tests. In the cognitive domain, the cluster 1 also had lower means for MOCA, and took longer to perform the cognitive TUG when compared to cluster 2.

### 3.4. Feature Selection

According to these data, age and the functional reach test are the most important variables for predicting road crash. When importance is assessed by the Mean Square Error (MSE), age is followed by campimetry overall, infractions, total work, and dominant side handgrip. On the other hand, when importance is evaluated by node purity, the functional reach test is followed by PT/BW (%), dominant side handgrip, non-dominant side handgrip, age, and TUG (Figure 2).

### 3.5. Observed vs. Predicted

Figure 3 shows the validation of the RF model with a high predictive capacity. The correlation between predicted and observed crashes was 0.98 and R^2^ was 0.81. Values ≤ 0.5 corresponded to the performance of a random model, values > 0.5 and < 0.6 indicate moderate predictive performance, and values > than 0.7 indicate good predictive performance [14]. 

## 4. Discussion

The main finding of the present study revealed that advanced age and the functional reach test were the two most important predictors of road crash, based on a car driving simulator with older drivers. Clustering methods are usually required during the early stages of knowledge discovery. We used the K-means algorithm, whose analysis approach was complementary to the RF model. K-means clustering identified similarities between individuals in the sample based on unsupervised learning where the algorithm itself identifies patterns. The RF model, which is based on supervised learning, that is, the learning process, involves both the variable that will be predicted and the predictors [25]. This algorithm was able to predict the number of crashes among older adult drivers and rank the main predictors. 

This study showed that changes related to the aging process in sociodemographic, physical, visual, and cognitive characteristics interfere with the ability to drive a vehicle in the older adult, evidencing the multimodal aspects of the ability to drive. Two main groups were identified in our sample based on the cluster analysis. Cluster 1 presented a longer time to brake the car, which may represent unsafe driving. Alonso et al. [11] demonstrated that older adults take 17% longer to brake the car than middle-aged adults; consequently, this increases the distance traveled before the vehicle stops. Congested areas and high speeds increase the probability of crashes [26]. However, there were no differences in the total number of crashes and infractions between the clusters.

In Cluster 1, when compared to Cluster 2, they had higher ages, driving times, and less education. The ability to drive declines with age, possibly explained by the age-dependent effect on visual, cognitive, and motor functioning that are needed to perform a complex task and maintain safe driving. Such functioning is found to decline at a high age [27], but this alone cannot be a factor that determines competence in driving, as aging is a heterogeneous process and affects individuals in different ways. 

It is important to mention that fewer years of schooling may present greater cognitive losses because according to [28], more years of schooling is a cognitive reserve and consequently, a protective factor. Therefore, more active, flexible, and resilient cognitive abilities can be seen as strategies to deal with everyday events, especially minimizing cognitive losses, typical of the aging process. 

Concerning the motor domain, cluster 1 presented lower handgrip strength (on both sides), shorter functional reach, shoulder flexion, and cervical rotation. This cluster also presented a long time to perform the TUG test. The aging process promotes a structural reorganization at the central and peripheral levels, which causes impairments in motor performance aggravated by deficiencies in the musculoskeletal system, including cellular and chemical alterations of the neuromuscular junction. These directly affect the pre- and postsynaptic, and consequently, voluntary muscle activation [29]. Considering all muscular action in the vehicular direction and with several joints involved, the motor performance will be affected and will suffer great functional impact in the face of the complexity of the task. Alonso et al. [11] demonstrated that the ability to take your foot off the accelerator and put it on the brake pedal in time to avoid collisions requires cognitive (central processing), sensory, and motor information. The decline of these functions increased braking time; consequently, we can infer that this increase leads to a greater number of crashes.

Regarding visual capacity, the drivers in Cluster 1, compared to the other group, presented lower values in the right, left, and binocular Snelling tests. In this sense, Chevalier et al. [30] affirm declines in the visual ability of older adults. They report that these individuals make shorter trips, stay closer to home, and drive less during the night.

The act of driving is dependent on vision [31]; most of the time incidents and collisions are attributed to vision deficit. Moreover, it influences the decision to stop driving in older adults. A problem in the visual field affects the transmission of information about surrounding vehicles, emergencies, obstacles, and mechanical problems in the vehicle itself. In addition to changes in visual acuity and field of vision, contrast sensitivity deficits are common in older adults, especially those diagnosed with cataracts, and are strongly associated with crashes. It is worse when both eyes are affected by cataracts. Intraocular lens placement reduced the risk of collision by 50% [32]. In our work, the visual assessment was not able to detect individuals who had cataracts, making it a limiting factor for a more accurate analysis of blindness.

According to the study conducted by Choi et al. [33], attention is fundamental for the efficiency of executive function in situations that demand driver´s readiness, and thus, it can quickly resolve conflicts in the face of associated tasks. Other studies also indicate that cognitive processing capacity is important for predicting driving performance, attributed to better performance in road tests and fewer safety failures [34,35]. Nonetheless in the present study, the cognitive domain, except for TUG Cognitive, did not differ between clusters, probably because people who volunteer for driving studies do not have significant cognitive, motor, or visual deficits, which may interfere with the responses during the driving simulator. However, studies that identify cognitive factors but do not consider physical limitations in older people may miss important clues. 

Although the present study evaluated different aspects related to the ability of driving, other factors that were not evaluated could help the K-Means algorithm to identify predictors and similarities between clusters. It could be, in a way, that there were no intersections and greater explanatory power on the occurrence of infractions and number of crashes in the driving simulator used.

The most important variables to predict road crash (using MSE) consisted of age campimetry overall, infractions, total work, and dominant side handgrip. However, when importance was evaluated by node purity, the functional reach test, PT/BW (%), dominant and non-dominant side handgrip, age, and TUG test were the most important features. During the training stage, we used 10-fold cross-validation, and this method was used to adjust the hyper parameters of the model. Regardless of the metric used to rank the predictors, the random forest model performed well in predicting the number of crashes among older adult drivers who were evaluated. Thus, among the set of cognitive, visual, and motor tests that can be used to assess the performance of driving a car safely considering the aging process, some predictors had a greater influence on the estimate and may be more sensitive than other aspects to foretell the number of crashes. Additionally, identifying and ranking these characteristics can guide which domains should be the object of intervention (e.g., when subject to physical, motor, visual, or cognitive rehabilitation) according to [6].

For the random forest model, in addition to the driver’s age, handgrip strength, visual ability, functional range test, and, to some extent, the assessment performed with the TUG could support assessment protocols in situations that simulate vehicular driving. The predictive model and the results presented in this study were based on a driving simulation in an older adult’s sample. Therefore, the order of importance of these factors may differ in other studies [11,16] or in real situations. 

Predictive samples using a driving simulator contribute to analyzing traffic safety and seeking information that is useful for preventive interventions. However, the literature on driving safety with older adults is very limited, as it is difficult to consider additional factors, for example, environment, road situation, traffic lights, etc. In the model that performed tests with a driving simulator associated with an artificial neural network, it showed that the driving of older adult drivers had a strong impact on the crash risk of following vehicles. They noticed that these individuals may increase the risk of collisions due to variations in speed in following cars, even if the older adult driver maintains a safe distance [36].

The main limitations of the study are related to the use of the driving simulator, precisely because it does not reproduce the real environment, with the influences acting at the same time, such as driving on the street, with the presence of people in different activities, other vehicles, pedestrians, and cyclists. However, the driving simulator allows situations that would be complex when compared to real conditions. In addition, the older adults in the present study did not present major cognitive, visual, and motor alterations, which opens perspectives for new studies in this area, such as musculoskeletal diseases, cognitive, and visual declines.

Second, Boyle and Lee [37] naturalistic studies bring information from critical safety events that are not easily identified in crash data. On the other hand, studies with car simulator studies are important to show underlying mechanisms related to the safety in vehicular driving such as the roads, driver, and vehicle characteristics that influence safety in vehicular driving. These two research approaches operate independently, but their integration can provide valuable insights.

The present study exposes intragroup similarities that could predict road crash and show the need for multifactorial assessments. The performance of several professionals, such as doctors, physiotherapists, occupational therapists, and psychologists, can interfere with the performance of the older adults, in addition to encouraging them to maintain the act of drive and consequently improve the quality of life of these individuals. Machine learning is a growing area for its potential to build predictive models with excellent performance. The present study opens a discussion on models that can identify groups of drivers whose characteristics are more likely to be involved in a crash and direct professionals and teams to which individuals could be evaluated more frequently, and in which domains the evaluations would require greater attention.

## 5. Conclusions

The results of two different algorithms used to assess motor, visual, and cognitive characteristics in older adult drivers were presented in our study. Although there were no differences in the number of crashes and infractions per cluster, the combination of models with supervised and unsupervised learning was able to identify subgroups of more similar individuals and predict the number of crashes with excellent accuracy, as well as rank the most important features. Data-driven decisions and machine learning can improve traffic safety and enable multi-professional teams to provide personalized assistance to older drivers.

## Figures and Tables

**Figure 1 ijerph-20-04212-f001:**
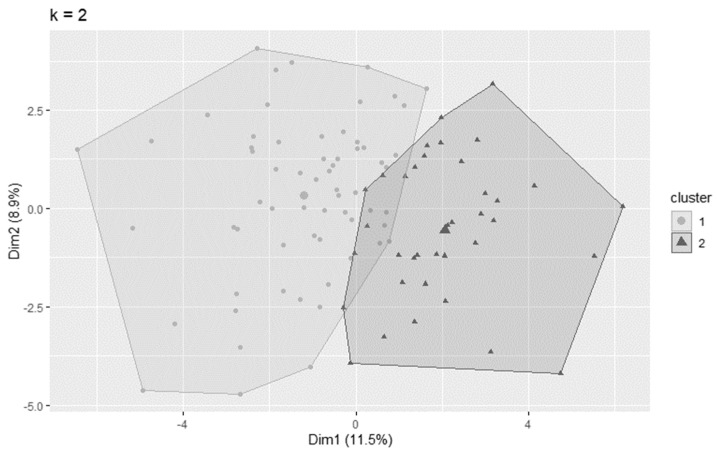
Cluster analysis of older adult drivers with the K—Means algorithm.

**Figure 2 ijerph-20-04212-f002:**
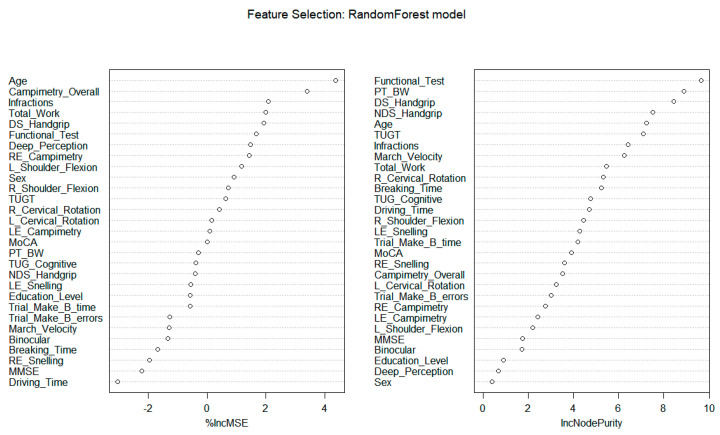
Importance of the variables to predict road crash through the Random Forest model. Legend: DS: Dominant Side; NDS: Non−Dominant Side; PT/BW: the peak of torque adjusted by body weight; TUGT: Timed Up and Go test; R: Right; L: Left; LE: Left Eye; RE: Right Eye; MoCA: Montreal Cognitive Assessment; Trails B: Trail Making Test Part B.

**Figure 3 ijerph-20-04212-f003:**
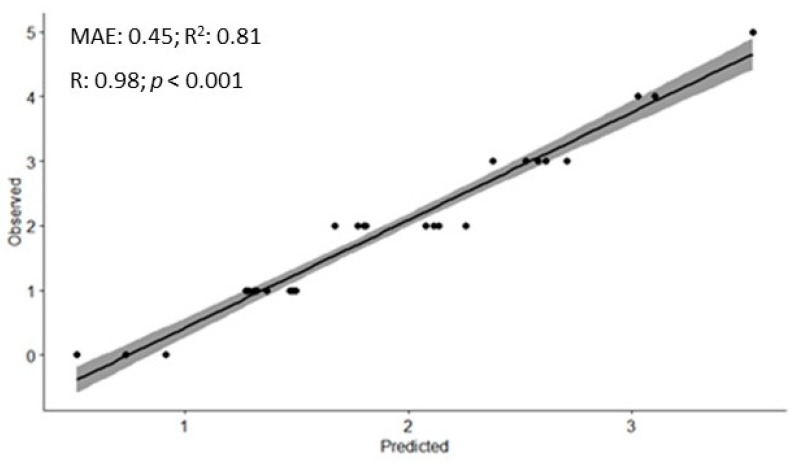
Validation of the random forest model for predicting road crash. Legend: MAE: Mean Absolute Error; R^2^: Coefficient of Determination; R: Correlation Coefficient.

**Table 1 ijerph-20-04212-t001:** Demographic and driving characteristics (n = 100).

Sociodemographic Data	Mean (SD)
Age (years)	72.5 (5.7)
Education Level (years)	12.3 (2.8)
Driving experience (years)	48.2 (7.0)
Road crash	1.8 (1.2)
Infractions	2.4 (2.1)
Braking Time (s)	0.95 (0.16)
Self-Perception of Difficulty	4.9 (3.0)
Legend: SD—standard deviation	

**Table 2 ijerph-20-04212-t002:** Sociodemographic data and motor, visual, and cognitive characteristics of older drivers by Cluster.

Sociodemographic	Cluster 1	Cluster 2	*p*-Value
	n = 59	n = 41	
Age (years)	74.9 (5.2)	69.2 (4.8)	<0.001 *
Education Level (years)	11.7 (2.8)	13.2 (2.6)	<0.05 *
Driving Time (years)	50.3 (6.8)	45.1 (6.2)	<0.05 *
No. Crashes	1.7 (1.1)	1.8 (1.1)	0.65
No. Infractions	2.6 (2.1)	2.0 (1.8)	0.14
Braking Time (ms)	0.98 (0.1)	0.89 (0.1)	<0.05 *
Motor domain			
DS Handgrip	29.9 (8.2)	34.9 (10.1)	<0.05 *
NDS Handgrip	27.1 (8.0)	31.8 (8.7)	<0.05 *
TUG (s)	9.2 (1.8)	7.9 (1.5)	<0.05 *
Reach Functional Test	31.2 (6.1)	33.7 (5.5)	<0.05 *
PT/BW plantar flexion (%)	67.5 (23.3)	93.2 (27.6)	<0.001 *
Total Work plantar flexion (J)	24.9 (0.3)	24.9 (0.3)	0.177
R Shoulder Flexion	159.4 (20.0)	167.2 (13.7)	<0.05 *
R Cervical Rotation	67.1 (11.9)	71.1 (10.2)	0.08
L Shoulder Flexion	157.9 (20.6)	167.5 (12.2)	<0.05 *
L Cervical Rotation	70.4 (18.0)	72.2 (12.1)	0.56
Visual domain			
RE Snelling	2.7 (2.2)	6.1 (3.2)	<0.001 *
LE Snelling	3.0 (2.4)	5.2 (3.2)	<0.05 *
Binocular	0.5 (0.8)	2.0 (1.4)	<0.001 *
RE Campimetry	86.0 (7.1)	85.6 (8.3)	0.79
LE Campimetry	85.6 (6.2)	87.1 (0.8)	0.20
Campimetry Overall	171.6 (12.0)	172.8 (111.8)	0.64
Cognitive domain			
MoCA	22.8 (3.6)	24.0 (3.2)	0.08
Trails B—errors	5.6 (5.8)	5.5 (7.3)	0.94
Trails B—time	159.3 (103.3)	147.0 (105.1)	0.56
TUG Cognitive	11.7 (2.9)	9.2 (2.6)	<0.001 *

Mean (SD). Legend: DS: Dominant Side; NDS: Non-Dominant Side; PT/BW: the peak of torque adjusted by body weight; TUG: Timed Up and Go test; R: Right; L: Left; LE: Left Eye; RE: Right Eye; MoCA: Montreal Cognitive Assessment; Trails B: Trail Making Test Part B. * *p* < 0.05.

## Data Availability

For reasons of confidentiality and privacy of the participants, the data used in this work are not available.

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
