# Peer review of "Crash Risk Predictors in Older Drivers: A Cross-Sectional Study Based on a Driving Simulator and Machine Learning Algorithms"

_ijerph, 2023, doi:10.3390/ijerph20054212_

Round 1

Reviewer 1 Report

This is a nice simple study that attempts to throw light on variables which may or may not impact on driver collisions. IIt uses novel methods of data analysis to try and sort the variables out to help understand what might be an issue in older driver behaviour. It does not find anything hugely signficant or world changing, but is a useful addition to the literature and to our knowledge on the subject.

Some minor changes needed:

- Consider changing two significant words:

Accident: Authors are beginning to be encouraged not to use the word "accident". Although it means ‘unintentional’, it is often interpreted as meaning ‘unavoidable’. More importantly, ‘accident’ is sometimes used to refer to the event (crash/collision/fall) and sometimes to the consequence (casualty/injury/fatality). It is not always clear which is meant. See BMJ 2001;322:1320 for a longer explanation.

Elderly: Please reconsider using the term “elderly” throughout the article and change it to “older people”, “older adults” or similar. See https://journals.lww.com/jgpt/fulltext/2011/10000/Use_of_the_Term__Elderly_.1.aspx for discussion on its use being ageist.

Also, more detail I think is needed on discussing the limitations of simulator use here, and its validity and reliability. How close is this similuator used here to real life? How close are the collisions to real life collisions and how can we tell? 

Finally, given the limitations, a little more on where next for the research? How might we get greater real life validity on these findings? How representative are the findings? How far can we generalise from them? 

Author Response

Review 1

Comments and Suggestions for Authors

This is a nice simple study that attempts to throw light on variables which may or may not impact on driver collisions. I It uses novel methods of data analysis to try and sort the variables out to help understand what might be an issue in older driver behaviour. It does not find anything hugely signficant or world changing, but is a useful addition to the literature and to our knowledge on the subject.

Some minor changes needed:
-Consider changing two significant words:
Accident: Authors are beginning to be encouraged not to use the word "accident". Although it means ‘unintentional’, it is often interpreted as meaning ‘unavoidable’. More importantly, ‘accident’ is sometimes used to refer to the event (crash/collision/fall) and sometimes to the consequence (casualty/injury/fatality). It is not always clear which is meant.

See BMJ 2001;322:1320 for a longer explanation.

Response: All accident words have been replaced by crash or crashes.

Elderly:  Please reconsider using the term “elderly” throughout the article and change it to “older people”, “older adults” or similar.

See https://journals.lww.com/jgpt/fulltext/2011/10000/Use_of_the_Term__Elderly_.1.aspx for discussion on its use being ageist.

Response: All the words elderly have been replaced by older adults

Also, more detail I think is needed on discussing the limitations of simulator use here, and its validity and reliability. How close is this similuator used here to real life? How close are the collisions to real life collisions and how can we tell? 

Finally, given the limitations, a little more on where next for the research? How might we get greater real life validity on these findings? How representative are the findings? How far can we generalise from them? 

Response: Second, Boyle e Lee, 2010, naturalistic studies brings information’s from critical safety events that are not easily identified in crash data. On the other hand, studies with car simulator studies are important to show underlying mechanisms related the safety in vehicular driving such as the roads, driver and vehicle characteristics influence safety in vehicular driving. These two research approaches operate independently, but their integration can provide valuable insights.

Reviewer 2 Report

The manuscript titled "Accident Risk Predictors in Older Drivers: A Cross-sectional Study Based on a Driving Simulator and Machine Learning Algorithms" is aimed to identify both motor, cognitive ,and visual predictors of car crashes as well as those variables that interfere with safe driving in a sample of older drivers. Using ML approaches, authors found 2 different clusters characterised by different age, driving time, and different braking time.

Although the topic is interesting I found hard to understand the paper in some points. I can say that these findings can be a relevant source of knowledge for those researchers who deal with traffic accident prevention and safe mobility in older drivers. Despite this, because of the above-mentioned reasons, I suggest major improvements.

Introduction

1) Authors employed visual, motor, and cognitive measures related to driving but essential visual, motor, and cognitive abilities for driving can be listed at least. They may explain more in depth what visual, motor, and cognitive measure (or test, or task) better discriminates safe/unsafe driving in older drivers. Since driving is mainly a visuospatial task, authors may consider expanding this concept by providing more information on the role that visuospatial cognitive abilities play in the driving performance of older:

- Anderson, S. W., Rizzo, M., Shi, Q., Uc, E. Y., & Dawson, J. D. 2005, June. Cognitive abilities related to driving performance in a simulator and crashing on the road. In Driving Assessment Conference (Vol. 3, No. 2005). University of Iowa;

- 10.3390/brainsci11081028;

- 10.1017/S1041610209009119;

- 10.1016/j.trf.2020.04.009;

- 10.3389/fpsyg.2020.604762;

- 10.1186/s40101-020-00227-9;

- 10.1093/ageing/29.6.517; 

- 10.1016/j.aap.2011.09.047; 

- Smyth J., Birrell S., Mouzakitis A., Jennings P. Motion sickness and human performance—Exploring the impact of driving simulator user trials; Proceedings of the 9th International Conference on Applied Human Factors and Ergonomics; Orlando, FL, USA. 21–25 July 2018)

 2) Authors deal with the predictors of traffic accidents in older drivers without providing any theoretical background on this topic. Authors could consider the “contextual-mediated model” of traffic accidents (10.1016/s0001-4575(02)00103-3) which is a widely accepted and referenced model.

1)      Line 72-73: Please, provide reference.

Statistical analysis

1)      Was a specific rule of thumb used to define the correct sample-size? If so, please, explain in depth procedures, parameters, and the used software.

2)      Line 204-205: Please, reduce the number of times in which you indicate the used software for statistical analysis.

3)      Was collinearity and multi-collinearity diagnostics performed? Since authors use Age, driving time (experience in years), and level of education among the set of predictors, they should control for potential multicollinearity. Please provide a correlation matrix (with partial correlations) as well as multicollinearity diagnostics (VIF and Tolerance).  

Results

Line 273-274: Please, explain what the finding means in terms of accuracy of the tested model as well as in terms of generalizability.

Discussion

Line 287: Please, remove the round bracket.

Author Response

Review 2

Comments and Suggestions for Authors

The manuscript titled "Accident Risk Predictors in Older Drivers: A Cross-sectional Study Based on a Driving Simulator and Machine Learning Algorithms" is aimed to identify both motor, cognitive, and visual predictors of car crashes as well as those variables that interfere with safe driving in a sample of older drivers. Using ML approaches, authors found 2 different clusters characterised by different age, driving time, and different braking time.

Although the topic is interesting I found hard to understand the paper in some points. I can say that these findings can be a relevant source of knowledge for those researchers who deal with traffic accident prevention and safe mobility in older drivers. Despite this, because of the above-mentioned reasons, I suggest major improvements.

Introduction
1) Authors employed visual, motor, and cognitive measures related to driving but essential visual, motor, and cognitive abilities for driving can be listed at least. They may explain more in depth what visual, motor, and cognitive measure (or test, or task) better discriminates safe/unsafe driving in older drivers. Since driving is mainly a visuospatial task, authors may consider expanding this concept by providing more information on the role that visuospatial cognitive abilities play in the driving performance of older:

- Anderson, S. W., Rizzo, M., Shi, Q., Uc, E. Y., & Dawson, J. D. 2005, June. Cognitive abilities related to driving performance in a simulator and crashing on the road. In Driving Assessment Conference (Vol. 3, No. 2005). University of Iowa;

- 10.3390/brainsci11081028;

- 10.1017/S1041610209009119;

- 10.1016/j.trf.2020.04.009;

- 10.3389/fpsyg.2020.604762;

- 10.1186/s40101-020-00227-9;

- 10.1093/ageing/29.6.517; 

- 10.1016/j.aap.2011.09.047; 

- Smyth J., Birrell S., Mouzakitis A., Jennings P. Motion sickness and human performance—Exploring the impact of driving simulator user trials; Proceedings of the 9th International Conference on Applied Human Factors and Ergonomics; Orlando, FL, USA. 21–25 July 2018). 

 2) Authors deal with the predictors of traffic accidents in older drivers without providing any theoretical background on this topic. Authors could consider the “contextual-mediated model” of traffic accidents (10.1016/s0001-4575(02)00103-3) which is a widely accepted and referenced model.

Response: Thank you for your suggestion. This study that you recommend (10.1016/s0001-4575(02)00103-3) it is related to personality and behavior, which is not the focus of our study. However, we add more information’s to support the introduction and included the following paragraph:

Driving is a complex task that involves the integration of motor, cognitive, and visual skills. The increase in age is related to changes in these capabilities that impact vehicle direction, which can contribute to the risk of a crash in older drivers. Age-related changes in visual function (mainly reduced visual acuity and visual field loss) have been investigated as risk factors for crashes among older adults [6]. According to [5], older drivers have more difficulty rapidly assessing and responding to roadway changes. Impaired contrast sensitivity was associated with a higher number of crashes and increased severity of crashes [7]. In addition, it is known that the natural cognitive decline of aging can affect driving, mainly due to attention deficit, impairments in memory, spatial and visual notions and a decline in attention are fundamental characteristics and directly impact safety while driving a vehicle [1]. The visuospatial skills involve the mental representations of the shape of objects and their locations and cognitive processes of transforming objects and movement through space. In the study by Anderson et al., [6] drivers who had accidents on the driving simulator showed worse performance in the composite measures of cognitive function, specifically in visuospatial skills and attention. Tinella et al. [7] evidenced significant results for the effect of global cognitive functioning on perceptual speed through the full mediation of mental rotation and perspective-taking skills. In another study by Tinella et al.,[8] the results suggest the specific contribution of spatial mental transformation skills in the execution of complex behaviors connected to the fitness to drive.

The physical condition of older adults explains the high risk of injuries and deaths. Alonso et al. [11] affirm muscle strength, postural balance, and cognition are associated with braking time and may affect driving performance in older adults. Age-related sensorimotor impairments also result in poor stepping reactions, altered patterns of movement coordination, and slower-reaching reactions in response to postural disturbances. Lower limb strength and control may be important for vehicle control, particularly in coordinating and adjusting the accelerator, brake, and clutch pedals. A test battery that assesses visual, cognitive, and sensorimotor domains are approach adopted by the clinical evaluation of drivers. However, there is still no consensus on which battery is the best, and what has been sought are tested with high predictive values with fewer variables.

  • Line 72-73: Please, provide reference.

Response: We add the reference.

Statistical analysis

  • Was a specific rule of thumb used to define the correct sample-size? If so, please, explain in depth procedures, parameters, and the used software.

Response: In this study was a convenience sample.

  • Line 204-205: Please, reduce the number of times in which you indicate the used software for statistical analysis.

Response: The name of the software was removed from these lines (204 e 205).

  • Was collinearity and multi-collinearity diagnostics performed? Since authors use Age, driving time (experience in years), and level of education among the set of predictors, they should control for potential multicollinearity. Please provide a correlation matrix (with partial correlations) as well as multicollinearity diagnostics (VIF and Tolerance).  

Supervised learning is a form of regression that relies on data input and output variables (heart and Gibney). Random Forest (RF) is a well-known and recognized supervised data mining technique. RF offers a means of efficiently modeling large and complex problems in which there may be hundreds of predictors. Variables that have many intersections. On the other hand, linear regression makes assumptions regarding the distributions of the data, homogeneity of variance, and that they are independent of one another (Smith, ganaeshi), Hingjun et al. (xx) have highlighted the success of Randon Florest in situations where regression assumption are often violated (i.e., a large number of predictors relative to sample size, multicollinearity, nonlinearity and higher order interactions between inputs.

Thus, despite our work not having focused on the multicollinearity of using predictors, we have taken such characteristics into account in the choice of the Random Forest as an analysis method.

Results

Line 273-274: Please, explain what the finding means in terms of accuracy of the tested model as well as in terms of generalizability.

Response:Values ≤ 0.5 corresponded to the performance of a random model, values > 0.5 and < 0.6 indicate moderate predictive performance and values > than 0.7 indicate good predictive performance (1).

Discussion

Line 287: Please, remove the round bracket.

Response: We did it. 

Thank you for your comments. 

Round 2

Reviewer 2 Report

Thank you for reviewing the manuscript which is notably improved in the current form. Before being ready for publication, however, I suggest the authors address the following issues: 

Response: In this study was a convenience sample.

Please add this information to the participant section. Also, report the number or percentage of female participants.

Response:Values ≤ 0.5 corresponded to the performance of a random model, values > 0.5 and < 0.6 indicate moderate predictive performance and values > than 0.7 indicate good predictive performance (1).

Please, Add these information to the manuscript.

Author Response

Thank you for reviewing the manuscript which is notably improved in the current form. Before being ready for publication, however, I suggest the authors address the following issues: 

Query 1: In this study was a convenience sample.

Please add this information to the participant section. Also, report the number or percentage of female participants.

Response: This information has been included

Line 110 - This is a study with a convenience sample of 100 older drivers...

Line 246-247 - 50(50%)female and 50(50%) male

Query 2: Values ≤ 0.5 corresponded to the performance of a random model, values > 0.5 and < 0.6 indicate moderate predictive performance and values > than 0.7 indicate good predictive performance (14).

Please, Add these information to the manuscript.

Response: This information has been included

Line: 290 a 293 - Values ≤ 0.5 corresponded to the performance of a random model, values > 0.5 and < 0.6 indicate moderate predictive performance and values > than 0.7 indicate good predictive performance [14].